**Data Availability Statement:** All relevant data are within the manuscript and its Supporting Information files.

# The diagnostic accuracy of the Mini-Cog screening tool for the detection of cognitive impairment—A systematic review and meta-analysis

Simisola Naomi Abayomi[1], Praveen Sritharan📍[2], Ellene Yan[3,4], Aparna Saripella[4], Yasmin Alhamdah[3,4], Marina Englesakis📍[5], Maria Carmela Tartaglia[3,6], David He[7], Frances Chung📍[3,4]*

1 Department of Medicine, University of Ottawa, Ottawa, Ontario, Canada, 2 Michael G DeGroote School of Medicine, McMaster University, Hamilton, Ontario, Canada, 3 Temerty Faculty of Medicine, Institute of Medical Science, University of Toronto, Toronto, Ontario, Canada, 4 Department of Anesthesia and Pain Management, Toronto Western Hospital, University Health Network, Toronto, Ontario, Canada, 5 Library & Information Services, University Health Network, Toronto, Ontario, Canada, 6 Temerty Faculty of Medicine, Division of Neurology, University of Toronto, Toronto, Ontario, Canada, 7 Department of Anesthesiology and Pain Medicine, Mount Sinai Hospital, University of Toronto, Toronto, Ontario, Canada

* frances.chung@uhn.ca

## Abstract

### Background

The Mini-Cog is a rapid screening tool that can be administered to older adults to detect cognitive impairment (CI); however, the accuracy of the Mini-Cog to detect CI for older patients in various healthcare settings is unclear.

### Objectives

To evaluate the diagnostic accuracy of the Mini-Cog to screen for cognitive impairment in older patients across different healthcare settings.

### Methods/Design

We searched nine electronic databases (including MEDLINE, Embase) from inception to January 2023. We included studies with patients ≥60 years old undergoing screening for cognitive impairment using the Mini-Cog across all healthcare settings. A cut-off of ≤ 2/5 was used to classify dementia, mild cognitive impairment (MCI), and cognitive impairment (defined as either MCI or dementia) across various settings. The diagnostic accuracy of the Mini-Cog was assessed against gold standard references such as the Diagnostic and Statistical Manual of Mental Disorders (DSM). A bivariate random-effects model was used to estimate accuracy and diagnostic ability. The risk of bias was assessed using QUADAS-2 criteria.

**Funding:** The author(s) received no specific funding for this work.

**Competing interests:** We have read the journal's policy and the authors of this manuscript have the following competing interests: Frances Chung reports research support from the Ontario Ministry of Health Innovation Grant, ResMed Foundation, University Health Network Foundation, Consultant to Takeda, and STOP-Bang Questionnaire proprietary to University Health Network. This does not alter our adherence to PLOS ONE policies on sharing data and materials.

**Abbreviations:** AUC, area under the curve; CERAD, Consortium to Establish a Registry for Alzheimer's Disease; CI, cognitive impairment; CDR, Clinical Dementia Rating; DOR, diagnostic odds ratio; DSM, Diagnostic and Statistical Manual of Mental Disorders; MCI, mild cognitive impairment; MMSE, Mini-Mental Status Examination; MoCA, Montreal Cognitive Assessment; NINCDS/ADRDA, National Institute of Neurological and Communicative Diseases and Stroke/Alzheimer's Disease and Related Disorders Association; NLR, negative likelihood ratio; NPV, negative predictive value; PLR, positive likelihood ratio; PPV, positive predictive value; SROC, summary receiver operating characteristic; QUADAS, quality assessment of diagnostic accuracy studies.

## Results

The systematic search resulted in 4,265 articles and 14 studies were included for analysis. To detect dementia (six studies, n = 4772), the Mini-Cog showed 76% sensitivity and 83% specificity. To detect MCI (two studies, n = 270), it showed 84% sensitivity and 79% specificity. To detect CI (eight studies, n = 2152), it had 67% sensitivity and 83% specificity. In the primary care setting, to detect either MCI, dementia, or CI (eight studies, n = 5620), the Mini-Cog demonstrated 73% sensitivity and 84% specificity. Within the secondary care setting (seven studies, n = 1499), the Mini-Cog to detect MCI, dementia or CI demonstrated 73% sensitivity and 76% specificity. A high or unclear risk of bias persisted in the patient selection and timing domain.

## Conclusions

The Mini-Cog is a quick and freely available screening tool and has high sensitivity and specificity to screen for CI in older adults across various healthcare settings. It is a practical screening tool for use in time-sensitive and resource-limited healthcare settings.

## 1. Introduction

Dementia or major neurocognitive disorder is defined as cognitive impairment in one or more cognitive domains severe enough to interfere with independent function [1]. Mild cognitive impairment (MCI) or mild neurocognitive disorder (NCD) is defined as cognitive impairment that minimally interferes with independent living [1]. The global pooled prevalence of undetected dementia in the community or residential area is very high at 61.7% [2]. Within the elective non-cardiac surgery setting, the pooled prevalence of unrecognized cognitive impairment (CI) is 37% and diagnosed CI is 18% [3]. CI is associated with an increased risk of prolonged length of hospital stay, and 1-year mortality [4,5].

A validated tool is needed to screen for CI (either MCI or dementia) across various healthcare settings given the high prevalence of unrecognized CI [6]. In fast-paced environments such as emergency units and preoperative clinics, screening tools may be practical for identifying at-risk individuals [7]. Early identification of CI can help with predicting outcomes, increased need for services, and prolonged recovery [8]. It can facilitate timely interventions aimed at disease-modifying therapy and maximize preparation for future care [9].

The Mini-Cog is a brief cognitive screening tool for dementia consisting of a three-item recall and a clock drawing task [10]. A common scoring method involves a score from zero to three for the three-item recall, corresponding to the number of correct recalls, and a binarized score for clock drawing as either normal or abnormal [11]. A total score of 0–2 from the clock drawing and 3-word recall tasks indicates high likelihood of CI. A total score of 3–5 indicates a low likelihood of CI [11]. Besides assessing older adults with dementia, the Mini-Cog may be useful to screen for cognitive impairment of varying severity including MCI, as the risk of MCI progression to dementia can occur at 11–33% within two years of diagnosis [12].

Previous systematic reviews have only explored the diagnostic accuracy of the Mini-Cog to detect only dementia across all age groups in primary [13,14] or secondary care settings [13–15]. MCI patients can be either amnestic or non-amnestic. Amnestic mild NCD commonly presents as a memory deficit whereas non-amnestic MCI usually presents as an impairment of executive function, attention, and visuospatial skills. The Mini-Cog consists of a clock drawing

and word recall test which allow for the assessment of these various domains that decline in those with MCI [12]. The diagnostic accuracy of the Mini-Cog to detect MCI in older adults has not been examined systematically. There is evolving interest in detecting cognitive impairment with varying severities in older adults [16,17]. This systematic review and meta-analysis aim to comprehensively evaluate the diagnostic accuracy of the Mini-Cog for assessment of CI (either MCI or dementia) in older adults across different healthcare settings.

## 2. Methods

### 2.1. Study registration

The study protocol was registered in the International Prospective Register of Systematic Reviews (PROSPERO) [CRD 42023396014]. We followed the Preferred Reporting Items for Systematic Reviews and Meta-Analyses (PRISMA) guidelines for the reporting of this systematic review (S4 Table) [18].

### 2.2. Search strategy

An information specialist (ME) performed a structured, systematic literature search. The following databases were searched from inception via the Ovid platform: MEDLINE, MEDLINE ePubs and In-Process Citations (daily), Embase, Cochrane Central Register of Controlled Trials, Cochrane Database of Systematic Reviews, and APA PsycINFO. The Web of Science Core Collection (Clarivate Analytics) and Scopus (Elsevier) were also searched. All the databases were searched on January 25, 2023 (Supplemental Material). Text word searching for "Mini-Cog" or "MiniCog" was conducted. Searches were limited to the English language. Conference materials were removed from results at source, where possible. To find papers citing the Mini-Cog, forward reference searching was conducted on the Web of Science Core Collection (Clarivate Analytics), Scopus (Elsevier), and CitationChaser (drawing on Lens.Org) [19] using one key article from the creators of the Mini-Cog tool [10]. Reference lists of included studies were examined by two reviewers (SA, PS) for articles missed in the initial search. The completed search strategies are provided in the supplementary material (S1 Table).

### 2.3. Study selection

Two reviewers (SA, PS) independently performed title, abstract, and full-text screening on Covidence according to the prespecified inclusion and exclusion criteria [20]. The inclusion criteria were: 1) older adults aged ≥ 60 years screened for CI (i.e. dementia or MCI) using the Mini-Cog; 2) the Mini-Cog was conducted in any healthcare settings, such as community, primary care, secondary care, emergency departments, and preoperative clinic; 3) cognitive status was compared against a reference standard, including the Diagnostic and Statistical Manual of Mental Disorders (DSM), the Petersen's Criteria, neuropsychological tests, and validated screening tests of the Mini-Mental State Examination (MMSE) and the Montreal Cognitive Assessment (MoCA); 4) test characteristics were reported, including sensitivity, specificity, positive predictive value (PPV), negative predictive value (NPV), and area under the curve (AUC); 5) cognitive status was blinded to the assessors of the Mini-Cog and the reference standard; and 6) randomized controlled trials and observational studies (case control, cross sectional, controlled trials, retrospective and prospective cohort). Conversely, studies were excluded if CI was not assessed using the validated Mini-Cog criteria or they were non-English language articles. All conflicts were resolved by a third reviewer (EY).

## 2.4. Data extraction

Following full-text screening, data extraction was completed independently by two reviewers (SA, PS), and discrepancies were resolved by a third reviewer (EY). The following information was extracted onto a standardized data collection sheet in Excel: authors, publication year, country, study design, healthcare setting, demographics, prevalence of CI, reference standards, and diagnostic accuracy of the Mini-Cog.

## 2.5. Quality of study assessment

The revised and validated version of the Quality Assessment of Diagnostic Accuracy Studies (QUADAS) tool was used to assess the quality of evidence provided by the included studies [21]. This tool assessed the studies across four domains: (1) patient selection, (2) index test, (3) reference standard, and (4) flow and timing. Each domain was scored as having "high", "low", or "unclear" risk of bias for each included study. For domains 1–3, an overall assessment of applicability was also provided for each study based on the bias assessment. This was completed by two reviewers (PS and YA), who independently rated the included studies, and conflicts were consolidated through discussion or by a third reviewer (EY). The inter-rater agreement was calculated for the overall risk of bias.

## 2.6. Statistical analysis

Qualitative and quantitative analyses were performed. For the qualitative analysis, study characteristics, demographics, and diagnostic accuracy were summarized in a tabular format for comparison. Studies were stratified based on the healthcare setting for analysis. We conducted the meta-analysis using Rev Man Review Manager version 5.4 and Meta disc version 1.4 [22,23]. A cut-off of two or less was used as the threshold for having dementia, MCI, and CI (defined as either dementia or MCI). A 2x2 contingency table was reconstructed for each outcome at a cut-off of two or less. Similar analysis was executed in different settings such as primary and secondary care. A random-effects bivariate analysis model was used to combine the results from each study and obtain the summary estimates with a 95% confidence interval (CI). Paired outcomes like the sensitivity and specificity of individual studies were analyzed using this method and a random-effects model was used to generate the forest plots (**S3 Table**). Test characteristics such as prevalence, sensitivity, specificity, PPV, NPV, log scale diagnostic odds ratio (DOR), and AUC were calculated with 95% CI (**S2 Fig**) [24]. A P-value of <0.05 was considered statistically significant. We used the $\chi 2$ test to explore the heterogeneity. The leave-one-study-out meta-analysis was performed to evaluate the effect of every study on the pooled estimates of sensitivity, specificity, log scale DOR, and heterogeneity. We performed meta-regression for the covariates age, female gender, and sample size using Open Meta Analyst [25]. The association of these covariates with the pooled estimates of sensitivity, specificity, and log scale DOR was assessed using meta-regression.

## 3. Results

### 3.1. Study selection and characteristics

The literature search resulted in 4,265 articles, with 1,193 remaining after removal of duplicates. After title and abstract screening, 1,125 articles were excluded, and 68 full-text articles were assessed for eligibility. Following full-text review, 14 articles (n = 7,194, mean age: 76.0 ± 7.7 years) were included in the study (**Fig 1**).

Of the 14 included studies, twelve articles were cross-sectional studies, and two were case-control studies. The countries of origin included the United States (n = 5), China (n = 2), Iran

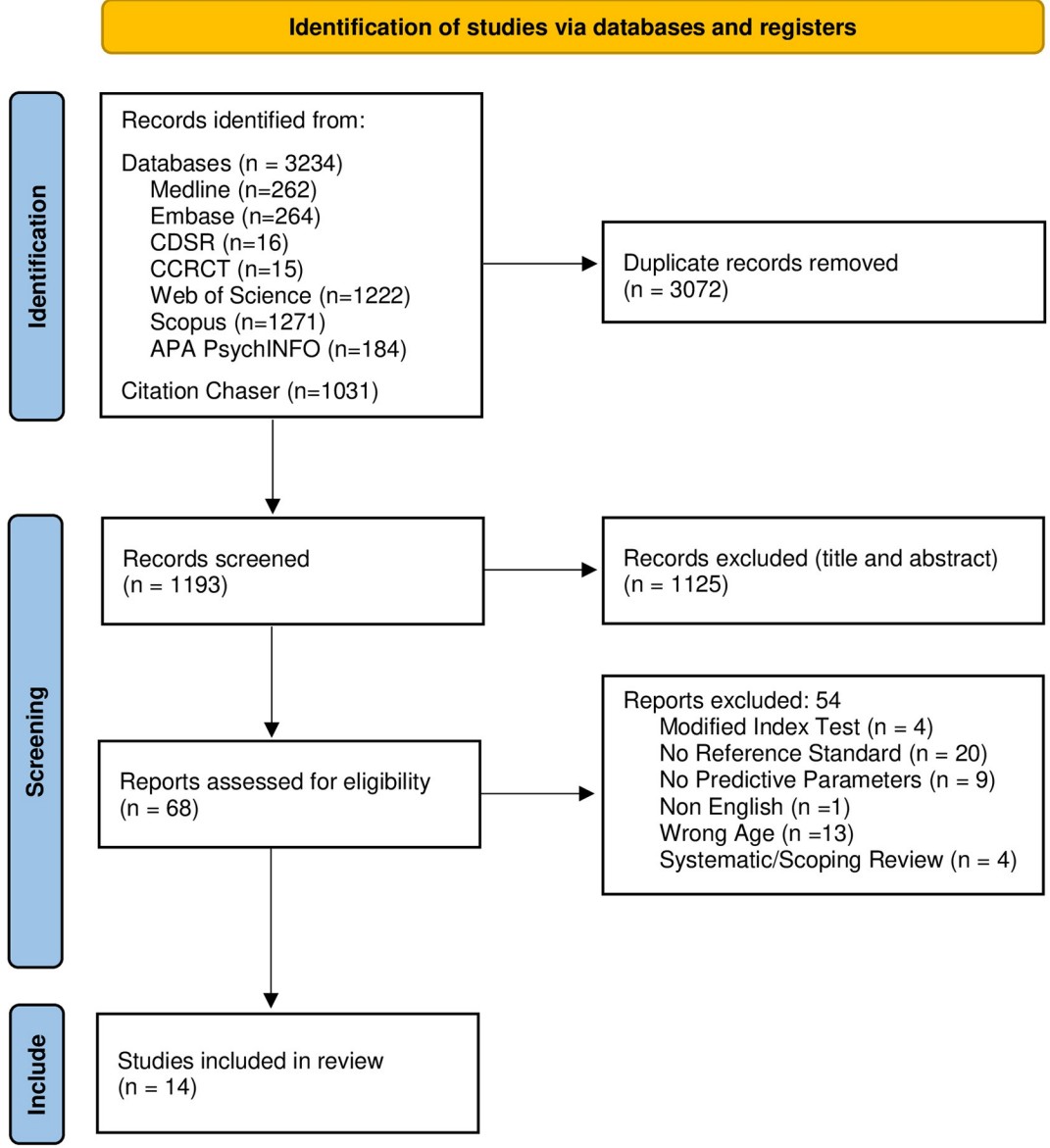

Abbreviations: APA, American Psychological Association; CCRCT, Cochrane Central Register of Controlled Trials; CDSR, Cochrane Database of Systematic Reviews.

**Fig 1. PRISMA flow diagram.**

(n = 2), Austria (n = 1), Belgium (n = 1), Brazil (n = 1), Portugal (n = 1) and Thailand (n = 1) (**Table 1**). We separated the included studies based on the healthcare settings: 1) community/ primary care [11,26–31], 2) secondary care/memory clinic [32–37] and 3) emergency department [38] (**Table 1**). Two of the included articles contained two datasets each, resulting in a total of 16 datasets analyzed in this systematic review and meta-analysis. We did not find any validation study of the Mini-Cog in the surgical population at the preoperative setting. We divided the included studies into three cognitive subgroups. The cognitive subgroups include: 1) dementia [26,28,31,35–37], 2) MCI [33,34], and 3) cognitive impairment (included either

**Table 1. Demographics and study characteristics of studies using Mini-Cog to detect dementia, mild cognitive impairment (MCI) and cognitive impairment (CI).**

| Author, year (country) | Patient population | Total No. of patients (N) | Age (years) mean ± SD | Sex (female) n (%) | Cognitive Status | Prevalence of Dementia/MCI/ CI N (%) | Reference standard(s) |
|---|---|---|---|---|---|---|---|
| *Community/primary care* | | | | | | | |
| **Borson 2003 USA** [26] | Community | 1119 | 73.1 ± 6.0 | 631 (54.6) | Dementia | 76 (6.8) | DSM-III-R, NINCDS-ADRDA and CDR. |
| **Borson 2005 USA** [11] | Community | 371 | Non-CI: 73.0 MCI: 74.0 Dementia 78.0 | 256 (69.0) | CI | 231 (62.2) | DSM-III-R, NINCDS-ADRDA and CDR. |
| **Costa 2012 Portugal** [27] | Community | 609 | 70.4 ± 6.8 | 368 (60.4) | CI | 19 (3.1) | MMSE <24 |
| **Holsinger 2012 USA** [28] | Primary care | 630 | 74.8 ± 6.6 | 45 (7.1) | CI | 269 (42.7) | DSM IV and NINCDS-ADRDA, Peterson/ Winbald criteria |
| **Holsinger 2012 USA** [28] | Primary care (veterans) | 630 | 74.8 ± 6.6 | 45 (7.1) | Dementia | 21 (3.3) | DSM-IV and NINCDS-ADRDA |
| **Kemenski 2009 Austria** [30] | Primary care | 86 | 77.5 ± 6.8 | 63 (72.0) | CI | 26 (30.0) | MMSE < 27 |
| **Shang 2021 China** [29] | Primary care | 160 | 80.7 ± 7.5 | 5 (3.1) | CI | 66 (41.3) | DSM V |
| **Yang 2016 China** [31] | Community | 2015 | 79.5 ± 7.6 | 1165 (57.8) | Dementia | 444 (22.0) | NIA-AA criteria |
| *Secondary care/memory clinic* | | | | | | | |
| **Clionksy 2010 USA** [37] | Neurology clinic | 702 | 78.2 ± 7.2 | 426 (61.0) | Dementia | 516 (73.5) | DSM IV |
| **Filho 2009 Brazil** [35] | Outpatient medicine clinic | 211 | 72.8 ± 5.4 | 153 (72.5) | Dementia | 91 (43.1) | DSM IV |
| **Ketelaars 2013 Belgium** [32] | Cancer clinic | 105 | 78.3 ± 5.1 | 51 (48.6) | CI | 31 (29.5) | MMSE < 24 |
| **Ketelaars 2013 Belgium** [32] | Geriatric clinic | 116 | 83.8 ± 4.7 | 81 (69.8) | CI | 77 (66.4) | MMSE < 24 |
| **Limpawattan 2021 Thailand** [33] | Outpatient internal medicine clinic | 150 | NR | 78(52.0) | MCI | 42 (28.0) | DSM V and geriatric psychiatrist evaluation |
| **Pourshams 2022 Iran** [34] | Geriatric and memory clinic | 120 | MCI 68.8 ± 8.0 Non-MCI 69.4 ± 7.4 | 62(52.0) | MCI | 60 (50.0) | DSM V and geriatric psychiatrist evaluation |
| **Rezaei 2018 Iran** [36] | Neurology clinic | 95 | D: 67.3 ± 6.1, ND: 65.8 ± 5.4 | 61 (64.0) | Dementia | 45 (47.9) | DSM V |
| *Emergency department* | | | | | | | |
| **Wilber 2015 USA** [38] | Emergency department | 75 | 75 ± 7.0 | 42 (56.0) | CI | 16 (21.0) | MMSE < 24 |

The Mini-Cog cut-off score was based on the original published algorithm (Borson et al. 2000) [10] which was translated into quantitative scale with possible scores of 0–5. Mini-Cog scores of 0–2 are in the impaired range and scores of 3–5 are in non-impaired range. A cut-off of ≥2/5 on the Mini-Cog was used to classify dementia, MCI, or CI (either MCI or dementia). Studies that detect cognitive impairment include patients with MCI or dementia. Classification of CI was defined by Borson 2005 [11] as "Dementia Probably AD," "Vascular Dementia," "Other Dementia," and "MCI."

Abbreviations: CDR, Clinical Dementia Rating; D, Dementia; DSM, Diagnostic and Statistical Manual of Mental Disorders; DSM-III-R, Diagnostic and Statistical Manual of Mental Disorders Third Edition Revised; MCI, Mild Cognitive Impairment; MMSE, The Mini Mental State Examination; ND, Non-Dementia; NIA-AA, National Institute on Aging-Alzheimer's Association; NINCDS-ADRDA, National Institute of Neurological and Communicative Disorders and Stroke and the Alzheimer's Disease and Related Disorders Association; NR, Not Reported.

dementia or MCI) [11,27–30,32,38] **(Table 1)**. Cognitive status was classified using the DSM (III-V) (n = 6) or more than one reference standard (n = 4). Cognitive status was also defined using the MMSE (cut-off score ranging from <23 to <27) (n = 4).

## 3.2. Pooled predictive parameters of Mini-Cog in different healthcare settings

In the community/primary-care setting, eight cross-sectional studies with 5,620 patients (mean age: 76.0 ± 7.7 years, 45.9% female) were included [11,26–31]. The overall prevalence of CI was 20% (95% CI: 19%-22%). The Mini-Cog showed 0.73 sensitivity (95% CI: 0.71–0.76, $I^2$: 97%) and 0.84 specificity (95% CI: 0.83–0.85, $I^2$: 94%). The AUC was 0.84 (95% CI: 0.76–0.89) (**Fig 2**, **Table 2**).

In the secondary care/memory clinic setting, seven studies with 1,499 patients (mean age: 76.2 ± 8.0 years, 60.8% female) were included. The overall prevalence of impaired cognition was 58% (95% CI: 55%-60%) [32–37]. The Mini-Cog showed 0.73 sensitivity (95% CI: 0.70–0.77, $I^2$: 93%) and 0.76 specificity (95% CI: 0.73–0.80, $I^2$: 86%). The AUC was 0.83 (95% CI: 0.74–0.89). (**Fig 2**, **Table 2**).

### Mini-Cog vs Reference Test in primary care/ community settings

| Study | TP | FP | FN | TN | Sensitivity (95% CI) | Specificity (95% CI) |
|---|---|---|---|---|---|---|
| Borson 2003 [26] | 58 | 115 | 18 | 928 | 0.76 [0.65, 0.85] | 0.89 [0.87, 0.91] |
| Borson 2005 [11] | 194 | 24 | 37 | 116 | 0.84 [0.79, 0.88] | 0.83 [0.76, 0.89] |
| Costa 2012 [27] | 7 | 59 | 12 | 531 | 0.37 [0.16, 0.62] | 0.90 [0.87, 0.92] |
| Holsinger 2012 C [28] | 16 | 164 | 5 | 445 | 0.76 [0.53, 0.92] | 0.73 [0.69, 0.77] |
| Holsinger 2012 PC [28] | 105 | 79 | 164 | 282 | 0.39 [0.33, 0.45] | 0.78 [0.73, 0.82] |
| Kemenski 2009 [30] | 20 | 24 | 6 | 36 | 0.77 [0.56, 0.91] | 0.60 [0.47, 0.72] |
| Shang 2021 [29] | 52 | 19 | 14 | 75 | 0.79 [0.67, 0.88] | 0.80 [0.70, 0.87] |
| Yang 2016 [31] | 391 | 236 | 53 | 1335 | 0.88 [0.85, 0.91] | 0.85 [0.83, 0.87] |

**Pooled Sensitivity = 0.73 (0.71 – 0.76); Chi-squared = 224.34; do = 7 (p= 0.0000); Inconsistency (I-square) = 96.9 %**
**Pooled Specificity = 0.84 (0.83 – 0.85); Chi-squared = 116.32; df = 7 (p= 0.0000); Inconsistency (I-square) = 94.0 %**
**AUC = 0.84 (0.76-0.89); SE (AUC) = 0.039**

### Mini-Cog vs Reference Test in secondary care/ memory clinic settings

| Study | TP | FP | FN | TN | Sensitivity (95% CI) | Specificity (95% CI) |
|---|---|---|---|---|---|---|
| Clionksy 2010 [37] | 346 | 24 | 170 | 162 | 0.67 [0.63, 0.71] | 0.87 [0.81, 0.92] |
| Filho 2009 [35] | 55 | 42 | 36 | 78 | 0.60 [0.50, 0.71] | 0.65 [0.56, 0.73] |
| Ketelaars 2013 C [32] | 25 | 12 | 6 | 62 | 0.81 [0.63, 0.93] | 0.84 [0.73, 0.91] |
| Ketelaars 2013 G [32] | 75 | 19 | 2 | 20 | 0.97 [0.91, 1.00] | 0.51 [0.35, 0.68] |
| Limpawattan 2021 [33] | 27 | 21 | 15 | 87 | 0.64 [0.48, 0.78] | 0.81 [0.72, 0.88] |
| Pourshams 2022 [34] | 59 | 14 | 1 | 46 | 0.98 [0.91, 1.00] | 0.77 [0.64, 0.87] |
| Rezaei 2018 [36] | 40 | 19 | 5 | 31 | 0.89 [0.76, 0.96] | 0.62 [0.47, 0.75] |

**Pooled Sensitivity = 0.73 (0.70– 0.76); Chi-squared = 88.45; df = 6(p= 0.0000); Inconsistency (I-square) = 93.2%**
**Pooled Specificity = 0.76 (0.73 – 0.80); Chi-squared = 41.63; df = 6 (p= 0.0000); Inconsistency (I-square) = 85.6 %**
**AUC = 0.83 (0.74-0.89); SE (AUC) = 0.035**

**Abbreviations: FN, false negative; FP, false positive; TN, true negative; TP, true positive. Ketelaars C, oncology geriatrics population; Ketelaars G, general geriatrics population; Holsinger C; dementia population; Holsinger PC, cognitive impairment population, AUC, Area under the curve, SE, standard error.**

**Fig 2. Pooled forest plots of sensitivity and specificity of Mini-Cog in each healthcare setting.**

In the emergency department, there was only one cross-sectional study with 75 patients (mean age: 75.0 ± 7.0 years, 21.0% female). The prevalence of impaired cognition was 21%. The Mini-Cog had a sensitivity of 0.75 (95% CI: 0.48–0.93) and a specificity of 0.85 (95% CI: 0.73–0.93) [38]. We did not find any validation study of Mini-Cog in the surgical population at the preoperative setting.

### 3.3 Pooled predictive parameters of Mini-Cog to detect dementia, MCI, and CI

The Mini-Cog was used in six studies (4,772 patients) across primary and secondary settings to detect dementia [26,28,31,35–37]. It showed 0.76 sensitivity (95% CI: 0.73–0.78, $I^2$: 94%) and 0.83 specificity (95% CI: 0.82–0.84, $I^2$: 95%) with an AUC of 0.85 (95% CI: 0.73–0.90) (**Fig 3**, **Table 2**).

In two studies (270 patients) at secondary/memory clinics, the Mini-Cog was used to detect MCI [33,34]. It showed 0.84 sensitivity (95% CI: 0.76–0.91, $I^2$: 96%) and 0.79 specificity (95% CI: 0.72–0.85, $I^2$: 0%). No summary receiving operating characteristic (ROC) plot was created due to two studies.

The Mini-Cog was used in eight studies (2,152 patients) across primary, secondary, and emergency department settings to detect CI (either MCI or dementia) [11,27–30,32,38]. It showed 0.67 sensitivity (95% CI: 0.63–0.70, $I^2$: 97%) and 0.83 specificity (95% CI: 0.81–0.85, $I^2$: 88%) with an AUC of 0.84 (95% CI: 0.75–0.87) (**Fig 3**, **Table 2**).

**Table 2. Pooled predictive parameters of the Mini-Cog based on cognitive status and healthcare settings.**

| Cognitive status | | | |
|---|---|---|---|
| **Predictive parameters** | **Dementia** | **MCI** | **CI** |
| | (6 studies, n = 4,772) [26,28,31,35–37] | (2 studies, n = 270) [33,34] | (8 studies, n = 2,152) [11,27–30,32,38] |
| Prevalence | 0.25 (0.24–0.26) | 0.38 (0.32–0.44) | 0.34 (0.32–0.36) |
| Sensitivity | 0.76 (0.73–0.78) | 0.84 (0.75–0.90) | 0.67 (0.63–0.70) |
| Specificity | 0.83 (0.82–0.84) | 0.79 (0.72–0.85) | 0.83 (0.81–0.85) |
| PPV | 0.60 (0.58–0.63) | 0.71 (0.62–0.79) | 0.68 (0.64–0.71) |
| NPV | 0.91 (0.90–0.92) | 0.89 (0.83–0.94) | 0.83 (0.81–0.85) |
| DOR | 15.67 (13.37–18.38) | 20.43 (10.66–39.15) | 10.98 (4.44–27.2) |
| SROC | AUC = 0.85 (0.73–0.90) | AUC = NA | AUC = 0.84 (0.75–0.87) |
| Healthcare settings | | | |
| **Predictive parameters** | **Primary/community care** | **Secondary care/ memory clinic** | **Emergency department** |
| | (8 studies, n = 5,620) [11,26–31] | (7 studies, n = 1,499) [32–37] | (1 study, n = 75) [38] |
| Prevalence | 0.20 (0.19–0.22) | 0.58 (0.55–0.60) | 0.21 |
| Sensitivity | 0.73 (0.71–0.76) | 0.73 (0.70–0.77) | 0.75 (0.48–0.93) |
| Specificity | 0.84 (0.83–0.85) | 0.76 (0.73–0.80) | 0.85 (0.73–0.93) |
| PPV | 0.54 (0.51–0.56) | 0.81 (0.78–0.83) | NA |
| NPV | 0.92 (0.92–0.93) | 0.67 (0.64–0.71) | NA |
| DOR | 14.20 (12.19–16.55) | 8.59 (6.78–10.88) | NA |
| SROC | AUC = 0.84 (0.76–0.89) | AUC = 0.83 (0.74–0.89) | NA |

Abbreviations: AUC, area under the curve; CI, cognitive impairment; DOR, diagnostic odds ratio; MCI, mild cognitive impairment; NA, not available; NPV, negative predictive value; PPV, positive predictive value; SROC, summary receiver operating characteristic. Studies that detect cognitive impairment include patients with MCI or dementia. Pooled predictive parameters based on healthcare setting detect cognitive impairment (defined as either dementia or MCI).

### Mini-Cog vs Reference test for dementia

| Study | TP | FP | FN | TN | Sensitivity (95% CI) | Specificity (95% CI) |
|---|---|---|---|---|---|---|
| Borson 2003 [26] | 58 | 115 | 18 | 928 | 0.76 [0.65, 0.85] | 0.89 [0.87, 0.91] |
| Clionksy 2010 [37] | 346 | 24 | 170 | 162 | 0.67 [0.63, 0.71] | 0.87 [0.81, 0.92] |
| Filho 2009 [35] | 55 | 42 | 36 | 78 | 0.60 [0.50, 0.71] | 0.65 [0.56, 0.73] |
| Holsinger 2012 [28] | 16 | 164 | 5 | 445 | 0.76 [0.53, 0.92] | 0.73 [0.69, 0.77] |
| Rezaei 2018 [36] | 40 | 19 | 5 | 31 | 0.89 [0.76, 0.96] | 0.62 [0.47, 0.75] |
| Yang 2016 [31] | 391 | 236 | 53 | 1335 | 0.88 [0.85, 0.91] | 0.85 [0.83, 0.87] |

**Pooled Sensitivity = 0.76 (0.73 – 0.78); Chi-squared = 77.84; df = 5 (p= 0.0000); Inconsistency (I-square) = 93.6 %**
**Pooled Specificity = 0.83 (0.82 – 0.84); Chi-squared = 108.64; df = 5 (p= 0.0000); Inconsistency (I-square) = 95.4 %**
**AUC = 0.85 (0.73-0.90); SE (AUC) = 0.038**

### Mini-Cog vs Reference test for MCI

| Study | TP | FP | FN | TN | Sensitivity (95% CI) | Specificity (95% CI) |
|---|---|---|---|---|---|---|
| Limpawattan 2021 [33] | 27 | 21 | 15 | 87 | 0.64 [0.48, 0.78] | 0.81 [0.72, 0.88] |
| Pourshams 2022 [34] | 59 | 14 | 1 | 46 | 0.98 [0.91, 1.00] | 0.77 [0.64, 0.87] |

**Pooled Sensitivity = 0.84 (0.76 – 0.91); Chi-squared = 23.70; df = 1 (p= 0.0000); Inconsistency (I-square) = 95.8%**
**Pooled Specificity = 0.79 (0.72 – 0.85); Chi-squared = 0.35; df = 1 (p= 0.5544); Inconsistency (I-square) = 0.0 %**

### Mini-Cog vs Reference test for CI

| Study | TP | FP | FN | TN | Sensitivity (95% CI) | Specificity (95% CI) |
|---|---|---|---|---|---|---|
| Borson 2005 [11] | 194 | 24 | 37 | 116 | 0.84 [0.79, 0.88] | 0.83 [0.76, 0.89] |
| Costa 2012 [27] | 7 | 59 | 12 | 531 | 0.37 [0.16, 0.62] | 0.90 [0.87, 0.92] |
| Holsinger 2012 [28] | 105 | 79 | 164 | 282 | 0.39 [0.33, 0.45] | 0.78 [0.73, 0.82] |
| Kemenski 2009 [30] | 20 | 24 | 6 | 36 | 0.77 [0.56, 0.91] | 0.60 [0.47, 0.72] |
| Ketelaars 2013 C [32] | 25 | 12 | 6 | 62 | 0.81 [0.63, 0.93] | 0.84 [0.73, 0.91] |
| Ketelaars 2013 G [32] | 75 | 19 | 2 | 20 | 0.97 [0.91, 1.00] | 0.51 [0.35, 0.68] |
| Shang 2021 [29] | 52 | 19 | 14 | 75 | 0.79 [0.67, 0.88] | 0.80 [0.70, 0.87] |
| Wilber 2015 [38] | 12 | 9 | 4 | 50 | 0.75 [0.48, 0.93] | 0.85 [0.73, 0.93] |

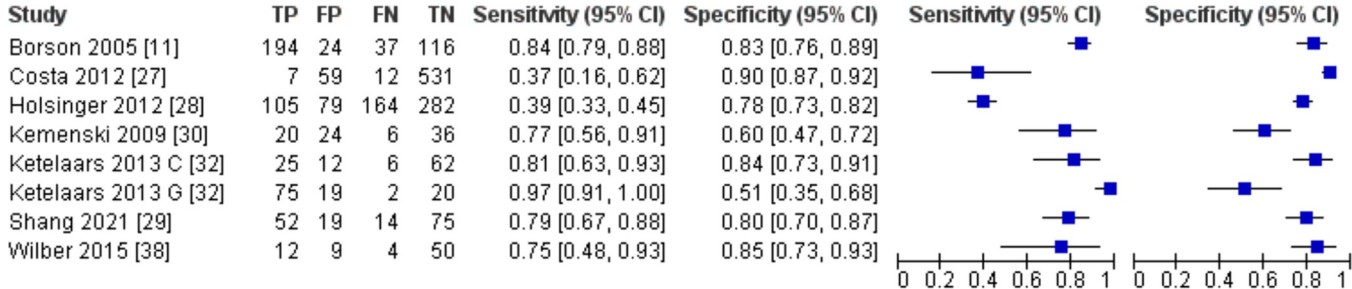

**Pooled Sensitivity = 0.67 (0.63 – 0.70); Chi-squared = 184.23; df = 7 (p= 0.0000); Inconsistency (I-square) = 96.2 %**
**Pooled Specificity = 0.83 (0.81 – 0.85); Chi-squared = 68.27; df = 7 (p= 0.0000); Inconsistency (I-square) = 89.7 %**
**AUC = 0.84 (0.75-0.87); SE(AUC) = 0.0295**

**Abbreviations: CI, cognitive impairment; FN, false negative; FP, false positive; MCI, mild cognitive impairment; NA, not available; TN, true negative; TP, true positive. Ketelaars C, oncology geriatrics population; Ketelaars G, general geriatrics population. Studies that detect cognitive impairment (CI) include patients with MCI or dementia.**

**Fig 3. Pooled forest plots of sensitivity and specificity of Mini-Cog based on cognitive status.**

## 3.4 Meta-regression and sensitivity analysis

The meta-analysis results showed the heterogeneity ($I^2$ value) to be greater than 50% for the calculated pooled predictive parameters in different stages of cognitive impairment and clinical settings. The meta-regression analysis was conducted to determine the influence of factors of the heterogeneity between studies on the pooled predictive parameters (**Tables 3 and 4**).

**Table 3. Meta-regression and sensitivity analysis of studies using the Mini-Cog to detect dementia.**

| Co-variate (Number of studies) | Sensitivity | | | Diagnostic Odds Ratio (Log scale) | | | Specificity | | |
|---|---|---|---|---|---|---|---|---|---|
| | Point Estimate (95% CI) | Coefficient (SE) | p-value | Point Estimate (95% CI) | Coefficient (SE) | p-value | Point Estimate (95% CI) | Coefficient (SE) | p-value |
| Age (6) | 0.773 (0.648–0.863) | -0.083 (0.032) | 0.008 | 2.58 (1.7–3.5) | -0.066 (0.070) | 0.346 | 0.791 (0.704–0.858) | 0.044 (0.066) | 0.502 |
| Female gender (6) | 0.773 (0.648–0.863) | 0.001 (0.002) | 0.651 | 2.58 (1.7–3.5) | 0.000 (0.002) | 0.995 | 0.791 (0.704–0.858) | 0.001 (0.001) | 0.563 |
| Sample Size (6) | 0.773 (0.648–0.863) | 0.001 (<0.001) | 0.099 | 2.58 (1.7–3.5) | 0.001 (0.001) | 0.241 | 0.791 (0.704–0.858) | -0.000 (<0.001) | 0.995 |

A moderate change in the combined estimates was observed following meta-regression and sensitivity analysis but this did not influence the total result. The meta-regression analysis showed that the confounders age and sample size were positively associated with CI detection using Mini-cog having a significant p value of < 0.05. No individual study significantly affected the outcome as illustrated by the leave one-study-out meta-analysis.

### 3.5 Risk of bias and applicability

The risk of bias as assessed through the QUADAS tool is displayed in the supplementary material (**S1 Fig, S2 Table**). Patient selection was found to have high risk of bias in 4/14 (28.6%) and unclear risk in 7/14 (50.0%) of studies. Seven (50.0%) studies were suspected of having high concerns of applicability regarding patient selection. Low ratings were primarily due to a lack of explanation regarding exclusions in patient samples and lack of clarity or avoidance of consecutive sampling methods to ensure robustness in patient selection. Flow and timing were found to have high risk in seven studies (50.0%) and unclear risk in two (14.3%). Low ratings were most given when studies excluded patients that were initially included without sufficient explanation. Both the index test and reference standard were assessed to have a low risk of bias and low levels of concerns regarding applicability in all 14 included studies. The inter-rater agreement was calculated on the risk of bias assessment and had a fair agreement of 80.5% (124/154 items) and a Cohen's Kappa of 0.23 (95% CI: 0.06, 0.41).

### 4. Discussion

In this systematic review and meta-analysis, we examined the diagnostic accuracy and predictive parameters of the Mini-Cog to detect CI in various healthcare settings such as primary care/community [11,26–31] secondary care/memory clinic [32–37] and emergency department [38]. We were unable to find any validation study in the preoperative setting for the

**Table 4. Meta-regression and sensitivity analysis of studies using the Mini-Cog to detect cognitive impairment (CI).**

| Co-variate (Number of studies) | Sensitivity | | | Diagnostic Odds Ratio (Log scale) | | | Specificity | | |
|---|---|---|---|---|---|---|---|---|---|
| | Point Estimate (95% CI) | Coefficient (SE) | p-value | Point Estimate (95% CI) | Coefficient (SE) | p-value | Point Estimate (95% CI) | Coefficient (SE) | p-value |
| Age (7) | 0.735 (0.512–0.880) | 0.275 (0.064) | <0.001 | 2.246 (1.4–3.1) | 0.163 (0.076) | 0.032 | 0.777 (0.669–0.858) | -0.127 (0.041) | 0.002 |
| Female gender (8) | 0.752 (0.547–0.884) | -0.003 (0.003) | 0.418 | 1.3 (0.82–1.76) | 0.000 (0.003) | 0.987 | 0.785 (0.695–0.854) | 0.003 (0.002) | 0.071 |
| Sample Size(8) | 0.752 (0.547–0.884) | -0.004 (0.001) | <0.001 | 1.3 (0.82–1.76) | -0.003 (0.001) | 0.022 | 0.785 (0.695–0.854) | 0.001 (<0.001) | 0.110 |

surgical population. To detect either MCI, dementia, or CI (MCI or dementia) in the primary care setting, the Mini-Cog yielded a pooled 73% sensitivity and 84% specificity. Within the secondary care, the Mini-Cog to detect MCI, dementia or CI (MCI or dementia) displayed a pooled 73% sensitivity and 76% specificity. To detect dementia, across primary and secondary care settings, the Mini-Cog showed 76% sensitivity and 83% specificity. To detect MCI, it had 84% sensitivity and 79% specificity. To detect CI (MCI or dementia), the Mini-Cog had 67% sensitivity and 83% specificity.

The Mini-Cog is an easy to administer and brief tool that is promising for application in primary and secondary care settings. Primary care providers are the initial contact for patients with neurocognitive decline [39]. They require rapid cognitive assessment tools to monitor cognitive changes from baseline or to inform timely referral to specialists [16,40]. Within secondary care/ memory clinics settings, cognitive screening tests are useful to identify participants who may require further diagnostic assessments. Comprehensive evaluations are costly and may not be practical for routine use in all patients [15]. The Mini-Cog is a free tool with a short administration time (3 minutes or less) that may be an appropriate alternative as a screening tool to comprehensive diagnostic tests.

We found that the prevalence of CI varies across different healthcare settings in this review. The prevalence of CI in the secondary/memory clinic was 58%, which is significantly higher than the prevalence of 20% in the primary care setting. Secondary care/memory clinic participants are likely given a referral for pre-existing signs of cognitive decline which may contribute to the elevated prevalence [41]. In a recent systematic review and meta-analysis of almost 7000 studies on the association of sensitivity and specificity with disease prevalence, a higher prevalence was associated with a higher estimated sensitivity and a lower estimated specificity [42]. This may explain why the specificity of the Mini-Cog test is lower in secondary care/ memory clinics compared to the primary/community care setting (0.76 vs 0.84). Clinicians should consider the implications of disease prevalence and spectrum when interpreting the results from studies of diagnostic test accuracy.

We found the diagnostic accuracy of the Mini-Cog to detect dementia, MCI, or CI to be similar in primary care/community and secondary care/ memory clinic settings. In the primary care/community setting, the pooled sensitivity and specificity of the Mini-Cog was 0.73 and 0.84 with an AUC of 0.86. Similarly in the secondary/memory clinic setting, the pooled sensitivity and specificity of the Mini-Cog was 0.73 and 0.76 with AUC of 0.84. In the emergency department setting, there was only one study and the reported sensitivity and specificity of the Mini-Cog was 0.75 and 0.85, respectively, comparable to the primary care setting [38].

Previous systematic reviews have examined the Mini-Cog in detecting only dementia, not MCI, nor CI (either MCI or dementia) in primary [13,14] and secondary care settings across all ages [13–15]. The limited number of included studies in the previous reviews was insufficient to determine the accuracy of the Mini-Cog to detect dementia [13–15].

The diagnostic accuracy of the Mini-Cog to detect MCI and CI in older adults has not been examined systematically. Our review evaluated the diagnostic accuracy of the Mini-Cog to detect CI which allowed for a comprehensive analysis of both dementia and MCI. Early identification of MCI is crucial for timely assessment in older adults who have not yet lost their independent function but may be at risk for further decline [43]. The Mini-Cog consists of a clock drawing and word recall test which allow for the assessment of various domains including episodic memory, visuospatial skills, and executive function that decline in those with MCI [33]. Screening for MCI may help prevent this decline by allowing healthcare providers to apply early prophylactic measures and treatment plans [44].

Currently the preferred tool to detect MCI is the MoCA, which has an excellent sensitivity of 0.79–0.90 and specificity of 0.70–0.81 compared to the MMSE (sensitivity 0.61–0.66,

specificity 0.65–0.74) [45]. Our review of the Mini-Cog revealed a sensitivity of 0.84 and a specificity of 0.79 to detect MCI. Compared with the MMSE, the Mini-Cog has superior diagnostic accuracy in detecting patients with MCI in a neurology outpatient setting, reporting an excellent sensitivity of 0.88 and specificity of 0.89 [17]. Due to the limitation of only two studies in the literature on the utilization of Mini-Cog to screen for MCI, further examination of the Mini-Cog is needed to determine the diagnostic accuracy of detecting MCI.

Currently, the MoCA and MMSE are screening tools commonly used to detect CI [46]. The limitations of these tools include a lengthy administration time of about ten minutes and copyright restrictions that require paid training for use [17]. In time-sensitive settings such as emergency departments, primary care clinics, or preoperative clinics, applications of these assessments may be limited [47,48]. The Mini-Cog is an accessible and freely available tool that takes less than five minutes to administer. It is less confounded by education compared to the MMSE and MoCA [49,50]. The validity of the Mini-Cog across different ethnic groups and degrees of cognitive decline allows for its applicability across several healthcare settings [17,51].

We did not find any validation studies on the Mini-Cog in the preoperative setting. There is a need to screen patients in this population for CI to inform appropriate perioperative management [3]. Pre-existing CI increases the risk of 1-month hospital readmission, discharge to assisted care, and postoperative delirium [5,52]. Positive screening for CI on the Mini-Cog has been associated with an increase in 3-month and 6-month postoperative mortality [53]. Given the high prevalence of unrecognized CI in older surgical patients, The Mini-Cog may help address the gap in identifying older adults with unrecognized CI at risk for postoperative complications [54]. Further validation is needed to evaluate the diagnostic accuracy of the Mini-Cog in the preoperative setting to determine if the Mini-Cog can predict postoperative outcomes such as delirium, morbidity, and mortality.

### 4.1 Limitations

There are various limitations to our systematic review and meta-analysis. Previous reviews have excluded case-control studies which were included in this study. The use of case-control studies introduces inherent sampling bias and the possibility for confounding variables to impact results. Additionally, various reference standards were used to evaluate the diagnostic accuracy of the Mini-Cog, which contributes to the heterogeneity of included studies. There were several factors that limited the number of studies included in our final analysis. This was due to unrecognized reference standards, variable cut-offs, and diverse patient demographics. Compared to dementia and CI subgroups, two studies detecting MCI were available, which impeded the calculation of AUC. Lastly, only one study was evaluated in the emergency department, which limited analysis in this setting.

## 5. Conclusion

The Mini-Cog is a brief screening test for CI in the older population that requires minimal training and resources. It has a high sensitivity and specificity to screen for CI in older adults across various healthcare settings. The test characteristics of the Mini-Cog make this a practical screening tool for use in time-sensitive healthcare settings. Future research is needed for utilization in preoperative assessment to validate the diagnostic accuracy of the Mini-Cog in the surgical population.

## Supporting information

**S1 Fig. Risk of bias and applicability graphs for QUADAS-2.**
(DOCX)

**S2 Fig. Summary receiver operating characteristic (SROC) of the Mini-Cog.**
(DOCX)

**S1 Table. Search strategy.**
(DOCX)

**S2 Table. QUADAS-2: Quality assessment of diagnostic accuracy studies.**
(DOCX)

**S3 Table. Predictive parameters of included studies using Mini-Cog to detect dementia, mild cognitive impairment (MCI) and cognitive impairment (CI).**
(DOCX)

**S4 Table. PRISMA checklist.**
(DOCX)

## Author Contributions

**Conceptualization:** Simisola Naomi Abayomi, Praveen Sritharan, Ellene Yan, Maria Carmela Tartaglia, David He, Frances Chung.

**Data curation:** Simisola Naomi Abayomi, Praveen Sritharan, Ellene Yan.

**Formal analysis:** Simisola Naomi Abayomi, Aparna Saripella, Yasmin Alhamdah.

**Methodology:** Marina Englesakis.

**Supervision:** Frances Chung.

**Validation:** Simisola Naomi Abayomi.

**Writing – original draft:** Simisola Naomi Abayomi, Praveen Sritharan, Ellene Yan, Aparna Saripella, Yasmin Alhamdah, Marina Englesakis, David He, Frances Chung.

**Writing – review & editing:** Simisola Naomi Abayomi, Praveen Sritharan, Ellene Yan, Aparna Saripella, Maria Carmela Tartaglia, David He, Frances Chung.

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
