## [Decision Letter · Decision Letter 0]

29 Nov 2023

PONE-D-23-31855The diagnostic accuracy of the Mini-Cog screening tool for the detection of cognitive impairment - A systematic review and meta-analysisPLOS ONE

Dear Dr. Chung,

Thank you for submitting your manuscript to PLOS ONE. After careful consideration, we feel that it has merit but does not fully meet PLOS ONE’s publication criteria as it currently stands. Therefore, we invite you to submit a revised version of the manuscript that addresses the points raised during the review process.

We look forward to receiving your revised manuscript.

Kind regards,

Ryota Sakurai, Ph.D.

Academic Editor

PLOS ONE

“I have read the journal's policy and the authors of this manuscript have the following

competing interests:Frances Chung reports research support from the Ontario Ministry of Health Innovation Grant, ResMed Foundation, University Health Network Foundation, Consultant to Takeda, and STOP-Bang Questionnaire proprietary to University Health Network.”

Additional Editor Comments:

Two reviewers suggested several critical concerns on this paper. Please respond to all these comments.

Reviewers' comments:

Reviewer's Responses to Questions

**Comments to the Author**

1. Is the manuscript technically sound, and do the data support the conclusions?

Reviewer #1: No

Reviewer #2: Partly

2. Has the statistical analysis been performed appropriately and rigorously? 

Reviewer #1: Yes

Reviewer #2: Yes

3. Have the authors made all data underlying the findings in their manuscript fully available?

Reviewer #1: No

Reviewer #2: Yes

4. Is the manuscript presented in an intelligible fashion and written in standard English?

Reviewer #1: Yes

Reviewer #2: Yes

5. Review Comments to the Author

Reviewer #1: Comments to the authors:

This systematic review and meta-analysis evaluated the diagnostic accuracy of the Mini-Cog tool to screen for cognitive impairment (CI) across various healthcare settings, mainly primary and secondary care settings. A qualitative analysis including the QUADAS-2 criteria for risk of bias assessment and an evidence table were used, and for quantitative analyses, the random-effects bivariate analysis of the included studies was conducted. The authors found that the sensitivity and specificity of the Mini-Cog to detect either MCI, dementia, or CI in primary care settings were 73% and 84%, respectively, whereas those of the secondary care settings were 73% and 76%, respectively. High and unclear risk of bias were found for patient selection and time intervals.

The merit of this study is the use of proper statistical methods for meta-analysis and a comprehensive qualitative analysis; however, this is highly hindered by the unclear definition and report of cognitive impairment based on the Mini-Cog score, which impedes the clinical feasibility and interpretation of the findings. Please kindly find below my points of concern that need to be addressed and some recommendations.

Overall:

L37-39, L61-62: The authors reported that the cut-off of ≤2/5 (according to the scoring rules published b Borson et. Al., 2000) was used to classify dementia, MCI, and CI (defined as either MCI or dementia)”; however, the authors are encouraged to report the cut-points used to distinguish dementia from MCI, especially, when they reported studies where MCI was detected (L45); otherwise, grouping MCI and Dementia as CI is quite misleading. Clarifying this will help not only enhance the potential value of this manuscript but also the applicability of the findings by helping clinicians make a more informed clinical decision.

For this reason, the authors are encouraged to redefine their purpose and definition of terms, improve the presentation of results and keep consistency of the report of the findings.

Introduction:

- L96-100: The authors indicate that previous reviews have reported the accuracy of the Mini-Cog to detect only dementia and imply that the diagnostic accuracy of the Mini-Cog to detect MCI or CI (either MCI or dementia). Please reconsider redefining your purpose or definition of term as commented above.

Results:

- PRISMA flow diagram: Please add the number of records identified from each database.

- Line 181 and Table 2: Although the authors reported that 14 studies were included for the final analysis, 15 studies are mentioned in Table 2, and Figures 2 and 3, and 16 studies are mentioned in Table 1. I assume this is because the authors reported two separate data from the same study, but this should be clearly reported in the Results section.

Discussion:

- Please report the findings in the first paragraph of this section.

- Table 2, for the Secondary care/memory clinic, the prevalence is higher than that of the primary/community care (0.58 vs 0.20), but also the specificity of the Mini-Cog test is lower than the primary/community care (0.76 vs 0.84). Do these values affect the prevalence in each healthcare setting? Please, kindly discuss these differences.

Tables and Figures:

- To ease readability, kindly add the reference number for each study in Tables 1 and 2 and Figures 2 and 3, just like it was reported in the main text (L193-194).

- Table 1: Please describe the Mini-Cog score used to define Dementia and MCI. As much as possible, I encourage authors to check the studies where they reported CI. For instance, Table 1 of the study of Borson et al., 2005 shows a classification of CI into “Dementia Probably AD,” “Vascular Dementia,” “Other Dementia,” and “MCI.”

- Figures 2 and 3: Please include the AUC values.

- Table 2: Please clearly indicate in the description that the values of the Healthcare settings are for CI in general; otherwise, please indicate the cognitive status reported in the studies for each healthcare setting.

Minor points:

- L135: A semicolon is needed between “(AUC)” and “5)”.

- L262: "setting" should be "settings"

- L202-203 and Table 2 of the Primary/community care, the AUC values are different. This also applies to the AUC and sensitivity values of the Secondary care/memory clinic.

- L213-217 and Table 2 of the Emergency department: Although in the main text, the authors report one study in the emergency department, in Table 2, this is reported as "NA". Please confirm.

Reviewer #2: This systematic review and meta-analysis evaluated the diagnostic accuracy of the Mini-Cog. I felt that the methods were rigorous overall. I did not find any crucial issues except for the area under the curve (AUC). I hope the authors find the following comments helpful for improving the manuscript.

Major Comments:

1. Introduction

Please clarify whether the Mini-Cog was initially developed to screen dementia and mild cognitive impairment. If my knowledge is correct, the cognitive test was developed for screening dementia. Please make clearer and justify further the theoretical rationale behind the extension beyond the original intended use.

2. Results

If possible, it would be better to conduct publication bias test for diagnostic accuracy of the Mini-Cog by funnel plot analysis.

3. Discussion

In this study, two diagnostic accuracy measures (i.e., sensitivity and specificity) were analyzed. Considering the inherent trade-off between sensitivity and specificity, it might be better to add AUC as the outcome measure. As Table 2 shows, however, the pooled AUC was not calculated in studies to detect mild cognitive impairment. At least, this issue should be acknowledged, as well as the small number of studies, in the Limitations subsection on page 18.

Minor Comments:

4. Results

If I did not overlook, please spell out abbreviations such as D and ND in Table 1. Also, the standard deviations of age seem lacking in Borson et al. (2005).

5. Results

Although Table 2 summarizes the pooled sensitivity and specificity, it would be better to present them in Figure 2 as well. If I understood correctly, the authors presented the sensitivity and specificity in each study only.

6. Results

If I understood correctly, S1 Table shows the results of meta-regression exploring the factors of the heterogeneity between studies. However, the descriptions on page 14 seem to be about the leave-one-study-out” meta-analysis. Please check and revise the text and supplementary materials if needed. Furthermore, if allowed, it would be better to present S1 Table in the main text.

7. Discussion

It might be better to refer to references #13 and #15 in the second paragraph in the Discussion section on page 15 and to clarify the contribution of this study. In the previous reviews, the number of studies was limited, and the evidence to recommend that the Mini-Cog should be used as a screening test for dementia was insufficient.

8. Discussion

Considering the limited number of studies to detect mild cognitive impairment by the Mini-Cog and the pooled sensitivity and specificity, I had a concern that the authors’ conclusion that the Mini-Cog is comparable to the MoCA on page 17 could be overemphasized. Further examinations would be needed to test whether the diagnostic accuracy of the Mini-Cog is comparable to the MoCA.

9. Discussion

When referring to the first limitation that this study included case-control studies, it would be better to mention that the previous reviews excluded case-control studies if I understood correctly.

6. PLOS authors have the option to publish the peer review history of their article (what does this mean?). If published, this will include your full peer review and any attached files.

Reviewer #1: No

Reviewer #2: No

---

## [Author Response · Author response to Decision Letter 0]

30 Dec 2023

Dear Dr. Ryota Sakurai and Reviewers, 

We appreciate your consideration, your valuable time, and insightful comments on our systematic review and meta-analysis “The diagnostic accuracy of the Mini-Cog screening tool for the detection of cognitive impairment - A systematic review and meta-analysis”. The comments have enabled us to make the manuscript much better. 

We have attached our specific response to reviewers as an additional document as a table format for ease. 

If there are any further comments, please do not hesitate to contact me at Frances.chung@uhn.ca. I appreciate your consideration of this manuscript and thank you for your time. 

Best regards,

Frances Chung MBBS MD 

Professor, Dept of Anesthesiology and Pain Medicine, University of Toronto,

ResMed Research Chair of Anesthesiology, Sleep, and Periop Medicine at University Health Network.

Clinical Investigator, Krembil Research Institute

Toronto Western Hospital, University Health Network

---

## [Decision Letter · Decision Letter 1]

4 Jan 2024

PONE-D-23-31855R1The diagnostic accuracy of the Mini-Cog screening tool for the detection of cognitive impairment - A systematic review and meta-analysisPLOS ONE

Dear Dr. Chung,

Thank you for submitting your manuscript to PLOS ONE. After careful consideration, we feel that it has merit but does not fully meet PLOS ONE’s publication criteria as it currently stands. Therefore, we invite you to submit a revised version of the manuscript that addresses the points raised during the review process. Two reviewers have commented on several points in the revised manuscript. Please respond to all these comments.

We look forward to receiving your revised manuscript.

Kind regards,

Ryota Sakurai, Ph.D.

Academic Editor

PLOS ONE

Journal Requirements:

Reviewers' comments:

Reviewer's Responses to Questions

**Comments to the Author**

1. If the authors have adequately addressed your comments raised in a previous round of review and you feel that this manuscript is now acceptable for publication, you may indicate that here to bypass the “Comments to the Author” section, enter your conflict of interest statement in the “Confidential to Editor” section, and submit your "Accept" recommendation.

Reviewer #1: All comments have been addressed

Reviewer #2: (No Response)

2. Is the manuscript technically sound, and do the data support the conclusions?

Reviewer #1: Yes

Reviewer #2: Yes

3. Has the statistical analysis been performed appropriately and rigorously? 

Reviewer #1: Yes

Reviewer #2: Yes

4. Have the authors made all data underlying the findings in their manuscript fully available?

Reviewer #1: Yes

Reviewer #2: Yes

5. Is the manuscript presented in an intelligible fashion and written in standard English?

Reviewer #1: Yes

Reviewer #2: Yes

6. Review Comments to the Author

Reviewer #1: Overall:

Thank you for addressing the suggested parts. There are some minor points to address. Please read below the comments and suggestions.

Abstract:

Line 31: From the authors’ response, “To date, there has been no systematic review and meta-analysis identifying the accuracy of the Mini-Cog to detect CI in different health care settings.”, please emphasize that the accuracy of the Mini-Cog to detect CI in different health care settings is unclear.

Lines 38-39 and 44-45: The sentences are repetitive. Following the authors’ response, please clarify in L44-45 that the cutoff was used across all settings. Please, make sure which symbol is appropriate “≤” (Line 38) or “≥” (Line 44).

Discussion:

Since the findings are being reported in the first paragraph of the Discussion section, please combine lines 280 and 281 together.

Reviewer #2: Thank you very much for revising the manuscript. I felt that the authors adequately addressed most of the comments raised in the previous review round. However, there appear to be some remaining issues. Below, I reiterate the previous comments. I hope the authors will find the following comments useful in improving the manuscript further.

Minor Comments:

1. Introduction:

To address the major comment #1, the authors referred to earlier studies using the Mini-Cog to screen mild cognitive impairment. However, it would be better to clarify the theoretical, not empirical, rationale behind the extension of the scope of the Mini-Cog. Specifically, how could the authors justify that the two cognitive tests (i.e., word recall and clock drawing) assess mild cognitive impairment?

2. Discussion:

To address the minor comment #9, the authors explained the limitation of the inclusion of case-control studies in the response letter in addition to the main text. However, if I understand the previous studies correctly, it would be fair to briefly explain that previous reviews excluded case-control studies and to clarify the weakness of the current review in the Limitations subsection. If my understanding is incorrect, the authors do not have to mention the previous reviews in the subsection.

7. PLOS authors have the option to publish the peer review history of their article (what does this mean?). If published, this will include your full peer review and any attached files.

Reviewer #1: No

Reviewer #2: No

---

## [Author Response · Author response to Decision Letter 1]

11 Jan 2024

Please see response to reviewers attached in files. Thank you for your consideration of this manuscript.

---

## [Decision Letter · Decision Letter 2]

23 Jan 2024

PONE-D-23-31855R2The diagnostic accuracy of the Mini-Cog screening tool for the detection of cognitive impairment - A systematic review and meta-analysisPLOS ONE

Dear Dr. Chung,

Thank you for submitting your manuscript to PLOS ONE. After careful consideration, we feel that it has merit but does not fully meet PLOS ONE’s publication criteria as it currently stands. Therefore, we invite you to submit a revised version of the manuscript that addresses the points raised during the review process.

We look forward to receiving your revised manuscript.

Kind regards,

Ryota Sakurai, Ph.D.

Academic Editor

PLOS ONE

Journal Requirements:

**Additional Editor Comments:**

The reviewer #1 gave very minor comments. Please respond to these.

Reviewers' comments:

Reviewer's Responses to Questions

**Comments to the Author**

1. If the authors have adequately addressed your comments raised in a previous round of review and you feel that this manuscript is now acceptable for publication, you may indicate that here to bypass the “Comments to the Author” section, enter your conflict of interest statement in the “Confidential to Editor” section, and submit your "Accept" recommendation.

Reviewer #1: All comments have been addressed

Reviewer #2: All comments have been addressed

2. Is the manuscript technically sound, and do the data support the conclusions?

Reviewer #1: Yes

Reviewer #2: Yes

3. Has the statistical analysis been performed appropriately and rigorously? 

Reviewer #1: Yes

Reviewer #2: Yes

4. Have the authors made all data underlying the findings in their manuscript fully available?

Reviewer #1: Yes

Reviewer #2: Yes

5. Is the manuscript presented in an intelligible fashion and written in standard English?

Reviewer #1: Yes

Reviewer #2: Yes

6. Review Comments to the Author

Reviewer #1: Thank you for revising the suggested parts.

I have some minor points to improve the readability. Other than this, I have no further comments.

Line 33: Please combine the sentences together using a transition word, such as "however" and punctuate it accordingly.

Line 102: Please define "NCD" at first mention.

Reviewer #2: Thank you very much for revising the manuscript further. I felt that the authors adequately addressed the remaining issues and that the revision clarified the theoretical rationale and the methodological rigor. Thus, I do not have any additional comments for improving the manuscript.

7. PLOS authors have the option to publish the peer review history of their article (what does this mean?). If published, this will include your full peer review and any attached files.

Reviewer #1: No

Reviewer #2: No

---

## [Author Response · Author response to Decision Letter 2]

25 Jan 2024

Minor changes were addressed. Thank you for your time.

---

## [Editor Report · Decision Letter 3]

30 Jan 2024

The diagnostic accuracy of the Mini-Cog screening tool for the detection of cognitive impairment - A systematic review and meta-analysis

PONE-D-23-31855R3

Dear Dr. Chung,

We’re pleased to inform you that your manuscript has been judged scientifically suitable for publication and will be formally accepted for publication once it meets all outstanding technical requirements.

Kind regards,

Ryota Sakurai, Ph.D.

Academic Editor

PLOS ONE
---

## [Editor Report · Acceptance letter]

5 Mar 2024

PONE-D-23-31855R3 

PLOS ONE

Dear Dr. Chung, 

I'm pleased to inform you that your manuscript has been deemed suitable for publication in PLOS ONE. Congratulations! Your manuscript is now being handed over to our production team.

Kind regards, 

on behalf of

Dr. Ryota Sakurai 

Academic Editor

PLOS ONE